# Supramolecular Fuzziness of Intracellular Liquid Droplets: Liquid–Liquid Phase Transitions, Membrane-Less Organelles, and Intrinsic Disorder

**DOI:** 10.3390/molecules24183265

**Published:** 2019-09-07

**Authors:** Vladimir N. Uversky

**Affiliations:** 1Department of Molecular Medicine and USF Health Byrd Alzheimer’s Research Institute, Morsani College of Medicine, University of South Florida, 12901 Bruce B. Downs Blvd. MDC07, Tampa, FL 33612, USA; vuversky@health.usf.edu; Tel.: +1-813-974-5816; Fax: +1-813-974-7357; 2Institute for Biological Instrumentation of the Russian Academy of Sciences, Pushchino 142290, Moscow Region, Russia

**Keywords:** intrinsically disordered protein, intrinsically disordered protein region, liquid–liquid phase transition, protein–protein interaction, protein–nucleic acid interaction, proteinaceous membrane-less organelle, fuzzy complex.

## Abstract

Cells are inhomogeneously crowded, possessing a wide range of intracellular liquid droplets abundantly present in the cytoplasm of eukaryotic and bacterial cells, in the mitochondrial matrix and nucleoplasm of eukaryotes, and in the chloroplast’s stroma of plant cells. These proteinaceous membrane-less organelles (PMLOs) not only represent a natural method of intracellular compartmentalization, which is crucial for successful execution of various biological functions, but also serve as important means for the processing of local information and rapid response to the fluctuations in environmental conditions. Since PMLOs, being complex macromolecular assemblages, possess many characteristic features of liquids, they represent highly dynamic (or fuzzy) protein–protein and/or protein–nucleic acid complexes. The biogenesis of PMLOs is controlled by specific intrinsically disordered proteins (IDPs) and hybrid proteins with ordered domains and intrinsically disordered protein regions (IDPRs), which, due to their highly dynamic structures and ability to facilitate multivalent interactions, serve as indispensable drivers of the biological liquid–liquid phase transitions (LLPTs) giving rise to PMLOs. In this article, the importance of the disorder-based supramolecular fuzziness for LLPTs and PMLO biogenesis is discussed.

## 1. Introduction to Proteinaceous Membrane-Less Organelles

It is recognized now that the cellular interior represents a highly crowded space, where various biological macromolecules (such as nucleic acids, polysaccharides, proteins, and ribonucleoproteins) occupy 5–40% of the cellular volume, and where the total concentration of these biological macromolecules can be as high as 80–400 mg/mL [1,2], with the total intracellular concentration of protein being expected to be up to 300 mg/mL, while the RNA levels can range from 20–100 mg/mL [3]. Importantly, recent studies revealed that all these biomacromolecules are distributed within a cell in a highly inhomogeneous manner, often forming different intracellular bodies or intracellular liquid droplets, which are known by different names, such as cellular (or nuclear) micro-domains, cellular (or nuclear, or mitochondrial) subdomains, intracellular (or intranuclear, or intramitochondrial, or intrachloroplast) bodies, non-membranous cytoplasmic (or nucleoplasmic) granules, and proteinaceous membrane-less organelles (PMLOs), which are commonly found in eukaryotic cells and bacteria [4,5,6,7,8,9,10,11,12]. Since PMLOs reversibly and controllably isolate target molecules in specialized compartments, they constitute an intricate answer to the cellular need to facilitate and control molecular interactions [5]. In fact, PMLOs serve as an important complement to the common membrane-encapsulated organelles, such as nucleus, mitochondria, Golgi apparatus, Golgi vesicles, smooth endoplasmic reticulum, rough endoplasmic reticulum, lysosomes, peroxisomes, secretory vesicles or granules (e.g., insulin granules), chloroplasts, and vacuoles. These membrane-bound organelles represent evolutionarily conserved compartments with complex barriers (membranes) permitting spatial isolation as well as energy-efficient and passive buffering of stochastic events [13].

Although traditional membrane-encapsulated organelles represent functionally optimized (and evolutionary conserved) compartments, where membranes provide the physical separation within a cell needed for some specialized processes to occur, PMLOs, which are also functionally optimized compartments, are not surrounded by a membrane (as follows from their name). PMLOs represent condensed heterogeneous liquid-like mixtures of proteins and nucleic acids formed via liquid–liquid phase separation (LLPS) or biological liquid–liquid phase transitions (LLPTs).

By concentrating specific proteins (and frequently RNA and/or DNA), biological LLPTs generate PMLOs, which are considered as intracellular functional hot spots that serve as organizers of cellular biochemistry [14,15]. The resulting PMLOs are many, and cytoplasmic granules include centrosomes [16], germline P-granules (germ cell granules or nuage) [17,18], neuronal RNA granules [19], processing bodies or P-bodies [20], and stress granules (SGs) [21]. There is only one type of PMLO in mitochondria and in chloroplasts, chloroplast SGs and mitochondrial RNA granules. On the other hand, the nucleus contains a large realm of nuclear PMLOs, such as nucleoli [22], nuclear pores [23], chromatin [24], Cajal bodies (CBs; [25]), nuclear stress bodies (nSBs) [26,27], nuclear gems (Gemini of coiled bodies or Gemini of Cajal bodies) [28,29], Sam68 nuclear bodies (SNBs) [30], perinucleolar compartment (PNC) [30], promyelocytic leukemia nuclear bodies (PML nuclear bodies) or PML oncogenic domains (PODs) [31], PcG bodies (polycomb bodies, subnuclear organelles containing polycomb group proteins) [32], paraspeckles [33], Oct1/PTF/transcription (OPT) domains [34], nuclear speckles or interchromatin granule clusters [35], histone locus bodies (HLBs) [36], and cleavage bodies [37], to name a few. This list represents only the tip of the iceberg, as recent studies suggest that 50+ different PMLOs can be found in eukaryotic cells and bacteria [4,5], and this number is increasing on a regular basis.

PMLOs are characterized by different physical properties, molecular compositions, subcellular localizations, cell type-specific features, and fast responsiveness to changes in cellular surroundings and environmental cues. In fact, PMLOs are dynamic, cell size-dependent, liquid-like bodies [9] with dimensions ranging from tens of nm to tens of μm and specific cellular distributions [11]. On the other hand, it has been shown that, although many intracellular bodies are liquid-like droplets with highly dynamic organization [10,15,38,39,40,41,42,43], some other PMLOs, e.g., amyloid bodies, centrosomes, nuclear pores, and Balbiani bodies, are characterized as “bioreactive gels” whose properties vary from solid-like to gels and viscous liquids [44]. Also, PMLOs are characterized by a high variability of their organizational complexity and compositions. In fact, based on their protein compositions (number of droplet-specific proteins), human PMLOs can be arranged in the following order: nucleolus (1626) > chromatin (1350) > nuclear speckles (650) > centrosome (530) > mitochondrial RNA granules (229) > promyelocytic leukemia protein (PML) nuclear bodies (104) > SGs (57) > perinuclear compartment (55) > Cajal bodies (CBs) (54) > polycomb group (PcG) bodies (48) > P-granules (Perinuclear RNA granules specific to the germline) (19) > nuage (18) > cleavage bodies (14) > Gemini (10) > SAM68 bodies (8) > paraspeckles (6) > nuclear SGs (5) = OPT (Oct1/PTF/transcription) domain (5) > histone locus bodies (HLBs) (4) = neuronal ribonucleoprotein (RNP) granules (4) [45]. Furthermore, environmental changes can also affect the protein composition and the physical properties of PMLOs [11], and this variability is controlled by different cellular factors, including (but not limited to) the stage of the cell cycle, the presence of growth stimuli, or stress [11].

LLPTs causing the PMLO formation may be triggered by a variety of environmental factors, such as: fluctuations in levels of biomacromolecules (proteins and nucleic acids) undergoing phase separation; variations in the concentrations of specific small molecules or salts; changes in temperature, osmolarity, and/or pH of the solution; various alterations of the phase-forming proteins caused by a multitude of posttranslational modifications (PTMs), alternative splicing, or binding of certain partners; or alterations of the environmental conditions modulating the protein–nucleic acid or the protein–protein interactions [8,9,14,46,47]. One should also keep in mind that the biological LLPTs and the related processes of PMLO formation are strongly condition-dependent, completely reversible, and tightly controlled [4,5]. This is schematically represented by Figure 1 showing LLPT and factors triggering these transitions.

Obviously, since there are no membranes around PMLOs, their biogenesis and structural coherence are exclusively governed by the intra-organelle protein–protein, protein–RNA, and/or protein–DNA interactions [48]. Furthermore, due to lack of surrounding membranes, the components of PMLOs are not protected from the environment and rapidly circulate between the organelle and its adjoining surroundings [49,50]. As a result, PMLOs exhibit several features characteristic of liquids. In fact, they show wettability (i.e., they can uphold contact with a solid surface) and possess sufficient surface tension for maintenance of their spherical shape. They can fuse upon contact, flow in response to shear stresses, and drip [17,21,51,52]. Therefore, based on their properties, PMLOs can be classified as a special liquid state (or liquid phase) of cytoplasm, matrix, nucleoplasm, or stroma characterized by the major physico-chemical properties that are rather close to the features of the corresponding intracellular fluids in which they are found [9]. On the other hand, although the intrinsic density and the viscosity of many PMLOs are relatively low, being not very different from those of the cytoplasm or the nucleoplasm [17,21,51,52,53,54,55,56], the PMLO interior is classified as an overcrowded milieu [4]. This is due to the fact that PMLOs contain noticeably higher total protein concentrations than those found within the crowded cytoplasm and the nucleoplasm [4]. An illustrative example of this overcrowded nature of PMLOs is given by nucleoli, speckles, and Cajal bodies of the *Xenopus* oocyte nucleus with the total protein concentrations of 215, 162, and 136 mg/mL, respectively. These values are noticeably higher than the total protein concentration of 106 mg/mL in the surrounding nucleoplasm [55]. More globally, although the dilute phase in a cell is maintained at the critical phase separation concentrations of proteins and nucleic acids [11], these biomacromolecules can be concentrated ~10–100-fold within the droplets [53,57], reaching millimolar concentrations [58].

Importantly, recent studies revealed that PMLOs are not homogeneous themselves. In fact, SGs were shown to be characterized by a heterogeneous structure, where the core was more densely packed and less easily accessible than the more diffused shell with easier exchange of the constituents between the SGs and the adjacent cytoplasm [59]. Because the components of a dense core are brought together at early stages of the SG assembly, whereas a diffused shell of these PMLOs is formed at later steps, these different SG phases are kinetically formed at different stages of the SG assembly [59]. Furthermore, using a combination of various in vivo and in vitro approaches with computational modeling, it was recently shown that one of the most studied PMLOs, the nucleolus, possesses layered droplet organization containing internal sub-compartments [60]. These sub-compartments were shown to represent distinct, coexisting, non-coalescing liquid phases formed by LLPTs of specific nucleolar proteins, suggesting that biological phase separation can generate multilayered liquids [60].

PMLOs are crucial for cellular functionality and are now considered as key organizers and regulators of many cellular processes [11]. Since multiple cellular components are concentrated within the PMLOs, they regulate a broad cohort of cellular processes ranging from transcription to translational repression, to RNP assembly and processing, to biogenesis of ribosomes, to transport and degradation of mRNA, and to intracellular signaling [15]. Because the LLPTs causing PMLOs are fast under normal physiological conditions, and because the PMLO components are concentrated in a dynamic, selective, and reversible manner, such intracellular bodies are well suited for processing of local information and for handling rapid and controllable responses to environmental alterations, indicating that at least some PMLOs can serve as dynamic sensors of localized signals [61].

Normally, the highly dynamic structure and composition of polyfunctional and multicomponent PMLOs allow them to provide finely tuned regulation of various intracellular processes. On the other hand, as with many other protein intrinsic disorder-based events and activities [62], even the slightest disruption of the activity of PMLOs and their biogenesis can lead to an imbalance of intracellular regulatory pathways, resulting in the development of various pathological conditions [40,63,64,65,66,67,68,69,70,71,72,73]. For example, although in their normal state, the majority of PMLOs (including SGs) possess liquid-like properties, their aging can promote development of a much less dynamic state that typically coincides with the appearance of fibrous structures [74]. Such aging-related alterations in the mechanical and the physical properties of PMLOs can be of biological and pathological significance [74]. For example, it was shown that the time-dependent changes in the dense core of aging SGs can promote formation of insoluble protein aggregates linked to neurodegenerative diseases [70,75].

## 2. Proteinaceous Membrane-Less Organelles, Liquid–liquid Phase Transitions, and Intrinsic Disorder

The facts presented in the previous section indicate that, typically, specific sets of resident proteins can be found in PMLOs. Among the characteristic properties uniting many of these PMLO-residing proteins is the presence of high intrinsic disorder levels, suggesting the overall importance of intrinsically disordered proteins (IDPs) or hybrid proteins with ordered domains and intrinsically disordered protein regions (IDPRs) for LLPTs and PMLOs [4,5,8,45,74,76,77,78,79,80,81,82,83,84]. In fact, the biogenesis of several PMLOs (e.g., nuages [57], nucleolus [85], P-granules [80], and RNA granules [74]) was shown to be critically dependent on IDPs/IDPRs. This is because the LLPTs driving the PMLO formation are determined by weak multivalent interactions between multi-domain proteins and/or IDPs, hybrid proteins with ordered domains and IDPRs [4,5,86], proteins with RNA-binding domains [87], proteins containing repeats of amino acids with polar and charged groups, or proteins with low complexity domains (LCDs) [5,9,88].

There are multiple reasons for why IDPs/IDPRs serve as the most appropriate candidates for biological LLPTs leading to PMLO formation. These reasons include: the overall high abundance of IDPs/IDPRs in various proteomes [89,90,91,92,93] {e.g., among the eukaryotic proteins, ~25–30% are mostly disordered [91], long IDPRs (longer than 30 residues) are found in more than half of eukaryotic proteins [89,90,91], whereas such long IDPRs are present in >70% of signaling proteins [94]}; their lack of fixed structure [95,96,97,98,99,100]; their high spatio-temporal heterogeneity and mosaic structural organization that constitute a mix of foldons, inducible foldons, morphing inducible foldons, non-foldons, semi-foldons, and unfoldons [86,100,101,102,103]; the ability of these proteins to serve as highly promiscuous binders engaged in a multitude of interactions with highly diversified partners and to thereby regulate and control a wide spectrum of cellular processes [95,97,98,99,100,104,105,106,107,108]; and their ability to preserve their mostly disordered status within PMLOs that defines the fluidity of these organelles and determines PMLOs as supramolecular fuzzy complexes (see below).

Weak, multivalent, and rather non-specific interactions between one or more IDPs/IDPRs and between IDPs/IDPRs and nucleic acids are expected to drive biological LLPTs, leading to the PMLO formation. The physico-chemical nature of these interactions driving phase separation can be highly diversified and range from π–π contacts to cation–π interactions [15], to hydrophobic interactions, and to heterologous and homologous electrostatic attraction between differently charged biological polymers and differently charged parts of the same protein molecules [4,5]. By virtue of the peculiarities of their amino acid sequences and biophysical properties, IDPs/IDPRs are uniquely positioned in the category of biological macromolecules capable of undergoing LLPTs and controlling the biogenesis of PMLOs. For example, the conformational behavior of IDPs/IDPRs is, at least in part, determined by the presence of a large number of charged residues and depletion in hydrophobic residues [95], which explains the mostly electrostatic nature of their interactions [109]. Since IDPs/IDPRs do not possess stable structures, existing in a form of highly dynamic conformational ensembles of rapidly interconverting flexible structures, mean electrostatic fields are created that are used in polyelectrostatic attraction [110]. Furthermore, since charged residues are typically heterogeneously distributed within the amino acid sequences of many IDPs/IDPRs, patches of similarly charged residues are generated, and such “block co-polymer”-like structure might serve as a good template for the electrostatics-driven LLPTs [5]. More generally, common presence in IDPs/IDPRs of arrays of tandem repeats with different physico-chemical properties [111] creates a foundation of the flexible multivalency needed for LLPTs [5]. Also, IDPs/IDPRs are known to be commonly subjected to various post-translational modifications (PTMs) [112,113]. As LLPTs can be regulated by PTMs [53], this PTM-controlled conformational and functional variability of IDPs/IDPRs is very appropriate for the regulation of PMLO biogenesis [5]. Being the “edge of chaos” systems [63,101,114,115], IDPs/IDPRs are known for their high sensitivity and responsiveness to (even rather subtle) environmental changes. Because of this environmental sensitivity and receptivity as well as the capability to undergo fast, highly controllable, environment-modulated transitions, IDPs/IDPRs play crucial roles in the regulation of LLPTs and PMLOs [5].

## 3. Dysregulation of the Biogenesis of Intracellular Liquid Droplets and Disease

It was pointed out that, since the local concentrations of proteins in PMLOs are noticeably higher than those in the surrounding crowded media (and, as a result, the interior of PMLOs is considered as the overcrowded milieu [4]), and since some amyloidogenic proteins can be found in PMLOs and many of these proteins can undergo LLPS both in vitro and in vivo, dysregulation of the biogenesis of intracellular liquid droplets can be related to various human diseases [63]. This suggests the existence of a definite spatio-temporal window of safe existence, where a given PMLO appears at a definite cell location in a response to a definite environmental cue and exists there for a definite amount of time, whereas the pathological conversion from liquid to solid or gel form within the highly concentrated milieu of PMLO might happen outside of this window of safe existence [40,63,64,67,69,70,71,72,74,75,79,116,117,118]. Generally, molecular mechanisms associated with the said pathological transformations are related to the dysregulated biogenesis of PMLOs, eventually leading to the distortions of their dynamics and the promotion of pathological aggregation. Some of these mechanisms include pathological “aging” of PMLOs (or going beyond the safe time window), increased content of proteins involved in LLPTs, aberrant PTMs, some chromosomal translocation, and pathological mutations [73]. Among proteins for which the aberrant LLPTs are associated with pathological aggregation are TAR DNA binding protein-43 (TDP-43) linked to amyotrophic lateral sclerosis (ALS) [116], microtubule-associated protein tau involved in Alzheimer’s disease (AD) [117,118], α-synuclein associated with Parkinson’s disease (PD) [66], TDP-43, heterogeneous nuclear ribonucleoprotein A1 (hnRNPA1) linked to ALS [66], fused in sarcoma (FUS) associated with the pathogenesis of ALS and frontotemporal lobar degeneration (FTLD) [119], prion protein [120], and many RNA-binding IDPs possessing low complexity domains (LCD) [64]. In other words, pathogenic transformations of PMLOs are often associated with the decreased fuzziness of these intracellular liquid droplets.

## 4. Supramolecular Fuzziness of Intracellular Liquid Droplets

An important feature of PMLOs is their fluidity. The liquid-like properties of phase-separated droplets facilitate the functions of their constituents, which are accumulated within droplets at high concentrations and show slowed diffusion but remain dynamic. In fact, the concentrations of proteins residing within these liquid droplets can be ~10–100-higher than the protein content of the dilute phase [53,57]. Furthermore, being intrinsically disordered, these PMLO-residing proteins can be engaged in multivalent interactions. These observations raise an important question regarding how fluidity can be preserved within the overcrowded milieu of the PMLO interior. It is likely that one can find an answer to this question by analyzing the structural properties of proteins within PMLOs or artificial phase-separated liquid droplets. In fact, if an IDP/IDPR would undergo global folding as a result of an LLPT, then the resulting condensed phase would not be liquid but would contain ordered protein–protein or protein–RNA complexes stabilized by multivalent rigid body–rigid body interactions. Therefore, the fact that PMLOs are liquid indicates that IDPs/IDPRs undergoing LLPTs preserve high levels of intrinsic disorder. Several recent NMR-based studies are in agreement with this hypothesis [121]. For example, the DEAD box protein 4 (DDX4), which is a probable ATP-dependent RNA helicase that serves as a primary constituent of nuage or germ granules [122], was shown by ^1^H-^15^N HSQC spectroscopy to remain disordered within the droplets [57]. Similarly, the LCD of the fused in sarcoma (FUS) protein, which is associated with two devastating neurodegenerative disorders—amyotrophic lateral sclerosis (ALS) and frontotemporal dementia (FTD) [123]—remained mostly disordered within the droplet phase [58]. Also, the microtubule associated protein tau, which is an IDP involved in Alzheimer’s disease [124,125] and other tauopathies [126], was shown to undergo LLPT in solution in a phosphorylation-dependent manner and preserved disordered state in the condensed phase [117]. Another IDP, BUB3-interacting and GLEBS motif-containing protein (BuGZ), the phase separation of which is involved in spindle matrix formation and function [127], was shown to remain dynamic in the spindle and its matrix [128].

The capability of IDPs/IDPRs to preserve high levels of disorder in their bound states is known as fuzziness, an important phenomenon emphasizing that formation, function, and/or regulation of the protein-based complexes/assembles are critically dependent on the intrinsic disorder of the constituent proteins [129,130]. Furthermore, it was emphasized that the biological activity of the resulting fuzzy complexes could be affected by fuzzy regions, which not only remained disordered but often escalated their conformational diversity in the bound state [131]. In fact, fuzzy regions are engaged in transient interactions, thereby establishing alternate contacts with specific partners. Flexibility and interactability of such regions can be regulated and controlled by PTMs and alternative splicing [131]. Because of the preservation of high disorder levels, PMLOs and artificial phase-separated liquid droplets represent fuzzy supramolecular complexes.

## 5. Conclusions

In summary, data accumulated to date indicate that high levels of intrinsic disorder are found in many PMLO resident proteins and show that the PMLO formation often relies on IDPs/IDPRs, indicating that PMLO biogenesis is crucially dependent on intrinsic disorder [8]. In other words, the lack of stable structure in IDPs/IDPRs, the ability of such proteins to be engaged in highly dynamic, weak, multivalent interactions combined with their capability to retain a highly mobile character after undergoing LLPTs define the liquid-like nature of PMLOs [5]. It is likely that the structural resilience of PMLOs and their capability to exist as stable entities in the absence of enclosing membranes combined with the free exchange of the constituents with the environment are also defined by the same properties of IDPs/IDPRs [5]. In summary, PMLOs are an enthralling form of disorder-based protein assemblages [4,5,86], which are formed without noticeable structural changes or ordering of their constituent IDPs/IDPRs when undergoing LLPTs, and which, as a result, are characterized by a highly dynamic nature defining their liquid-like appearance [57]. In other words, supramolecular fuzziness is crucial for many aspects of PMLO biogenesis, stability, and functionality.

## Figures and Tables

**Figure 1 molecules-24-03265-f001:**
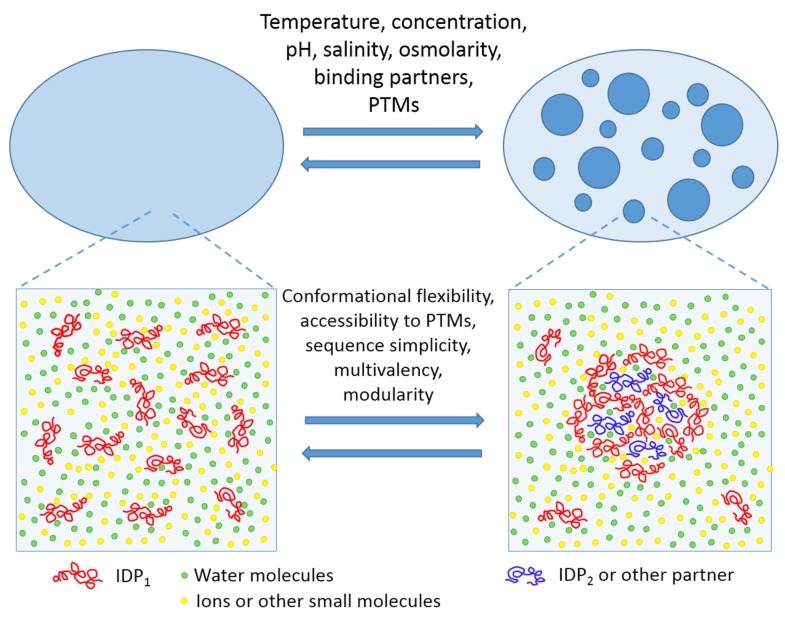
Schematic representation of liquid–liquid phase transitions (LLPT) and thermodynamic factors (top) and intrinsic disorder-related features controlling liquid–liquid phase transitions in protein solutions. This figure is reprinted from Current Opinion in Structural Biology, Vol. 44, Uversky V.N. Intrinsically disordered proteins in overcrowded milieu: membrane-less organelles, phase separation, and intrinsic disorder, Pages No. 18–30, Copyright 2017, with permission from Elsevier.

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
