# Peer review of "Supramolecular Fuzziness of Intracellular Liquid Droplets: Liquid–Liquid Phase Transitions, Membrane-Less Organelles, and Intrinsic Disorder"

_molecules, 2019, doi:10.3390/molecules24183265_

Round 1

Reviewer 1 Report

The perspective article on "supramolecular fuzziness of intracellular liquid droplets" covers a very timely subject because our awareness of the variety, composition, formation, functions and importance of these "membrane-less organelles" has been growing rapidly in recent years.  The review gives a helpful description of the various types and their locations and a good, clear explanation of our current understanding of their composition, functions, and other properties.  I have just a few suggesitons:

Addition of a figure or table would be helpful, perhaps a table of examples, their cellular locations, and functions 

The section containing lines 58-67 about factors affecting formation would be better after the paragraphs about the specific types, so after line 102.

remove "the" in line 15, line 15 again, 19, 57, 59, 156, 2 places in 186, 2 places in 187, before differently in 188, 199, 251, 253

line 56 - a membrane

62 - a multitude

69 -considered to be

74 the nucleus contains

76 remove 2nd of?, and remove the before Sam 68

84 compositions

87 distributions

105 lack of surrounding membranes

128 Because the components of a dense core are brought together at the early states of the SG assembly, whereas a diffuse shell of these PMLOs is formed

143 well suited

144 responses

151 the majority

190 unique position in the

193 interactions

205 high sensitivity and

206 because of this

208 in the regulation

233 in the spindle

239 conformational diversity

249 retain a highly mobile

253 In summary,

255 when undergoing

Author Response

The perspective article on "supramolecular fuzziness of intracellular liquid droplets" covers a very timely subject because our awareness of the variety, composition, formation, functions and importance of these "membrane-less organelles" has been growing rapidly in recent years.  The review gives a helpful description of the various types and their locations and a good, clear explanation of our current understanding of their composition, functions, and other properties. 

REPLY: Thank you for careful reading of the manuscript, helpful suggestions and high evaluation of this work.

I have just a few suggestions:

Addition of a figure or table would be helpful, perhaps a table of examples, their cellular locations, and functions

REPLY: To address this suggestion, a figure was added showing LLPT and factors triggering this transition.

The section containing lines 58-67 about factors affecting formation would be better after the paragraphs about the specific types, so after line 102.

REPLY: Corrected

remove "the" in line 15, line 15 again, 19, 57, 59, 156, 2 places in 186, 2 places in 187, before differently in 188, 199, 251, 253

REPLY: Corrected

line 56 - a membrane

REPLY: Corrected

62 - a multitude

REPLY: Corrected

69 -considered to be

REPLY: Corrected

74 the nucleus contains

REPLY: Corrected

76 remove 2nd of?, and remove the before Sam 68

REPLY: Corrected

84 compositions

REPLY: Corrected

87 distributions

REPLY: Corrected

105 lack of surrounding membranes

REPLY: Corrected

128 Because the components of a dense core are brought together at the early states of the SG assembly, whereas a diffuse shell of these PMLOs is formed

REPLY: Corrected

143 well suited

REPLY: Corrected

144 responses

REPLY: Corrected

151 the majority

REPLY: Corrected

190 unique position in the

REPLY: Corrected

193 interactions

REPLY: Corrected

205 high sensitivity and

REPLY: Corrected

206 because of this

REPLY: Corrected

208 in the regulation

REPLY: Corrected

233 in the spindle

REPLY: Corrected

239 conformational diversity

REPLY: Corrected

249 retain a highly mobile

REPLY: Corrected

253 In summary,

REPLY: Corrected

255 when undergoing

REPLY: Corrected

Reviewer 2 Report

The article is well written, treat an interesting issue, reads well and is logically sound. Since the manuscript type is perspective, physical chemistry section some questions should be addressed to improve the potential readers.

Chapter 1, row 53 to 63: the author discusses “LLPTs causing the PMLO formation may be triggered by a variety of environmental factors”. At the end of this discussion, the author should introduce the physical chemistry of this behavior i.e. Chemical equilibrium between aggregate and monomeric molecules that is the basis of PMLO formation and how changes in thermodynamic coordinate change the equilibrium.

Chapter 1, row 109: “…and possess sufficient surface tension for maintenance of their spherical shape.” Discuss the calculation of Gibbs free energy of PMLO since surface tension multiplied to SASA has done delta G and not only to the morphology.

Chapter 2. In this paragraph, IDSs behavior is discussed. Any discussion about amyloidogenic proteins is done. This class of proteins covers a very important field of IDPs since many pathologies are involved (see type II diabetes, Parkinson and Alzheimer diseases). Moreover, the physical chemistry of amyloidogenic proteins has contributed to increasing our knowledge about the structure and dynamics of amyloid and amyloidogenic proteins. Amyloidogenic proteins form soluble unstructured transient aggregates that are responsible for the toxicity of some IDPs such as human IAPP, beta-amyloid, and alpha-synuclein. The LLPTs approach could give a new impulse to understanding the molecular mechanisms of their toxicity. Besides, it has recently been reported that IDPs form stable aggregates in the liquid phase even with free phospholipids in solution (10.1063/1.4948323; 10.1021/acs.jpclett.8b02241).

Last research can be find on the following papers: doi:10.1021/ja900285z; 10.1021/jp511758w; 10.1038/srep28658; 10.1021/ja405993r; 10.1021/acschemneuro.7b00110.

Last, please discuss G-quadruplexes another interesting class of PLMOs molecules (some information can be found on DOI: 10.1016/j.bmc.2013.12.051; 10.1016/j.tibtech.2017.06.012)

Author Response

The article is well written, treat an interesting issue, reads well and is logically sound. Since the manuscript type is perspective, physical chemistry section some questions should be addressed to improve the potential readers.

REPLY: Thank you for careful reading of the manuscript, helpful suggestions and high evaluation of this work.

Chapter 1, row 53 to 63: the author discusses “LLPTs causing the PMLO formation may be triggered by a variety of environmental factors”. At the end of this discussion, the author should introduce the physical chemistry of this behavior i.e. Chemical equilibrium between aggregate and monomeric molecules that is the basis of PMLO formation and how changes in thermodynamic coordinate change the equilibrium.

REPLY: To address this issue, a figure was added showing LLPT and factors triggering this transition.  

Chapter 1, row 109: “…and possess sufficient surface tension for maintenance of their spherical shape.” Discuss the calculation of Gibbs free energy of PMLO since surface tension multiplied to SASA has done delta G and not only to the morphology.

REPLY: I am not sure that addition of this discussion is needed. In my view, this discussion is unrelated to the subject of this perspective.

Chapter 2. In this paragraph, IDSs behavior is discussed. Any discussion about amyloidogenic proteins is done. This class of proteins covers a very important field of IDPs since many pathologies are involved (see type II diabetes, Parkinson and Alzheimer diseases). Moreover, the physical chemistry of amyloidogenic proteins has contributed to increasing our knowledge about the structure and dynamics of amyloid and amyloidogenic proteins. Amyloidogenic proteins form soluble unstructured transient aggregates that are responsible for the toxicity of some IDPs such as human IAPP, beta-amyloid, and alpha-synuclein. The LLPTs approach could give a new impulse to understanding the molecular mechanisms of their toxicity.

REPLY: Although formation of oligomers by amyloidogenic IDPs is not related to the subject of this article, discussion of the “aging” of PMLOs, their pathological transformation and potential relation of these processes to the pathogenesis of human diseases was added to the revised manuscript as a new section entitled “Dysregulation of the biogenesis of intracellular liquid droplets and disease”.

Besides, it has recently been reported that IDPs form stable aggregates in the liquid phase even with free phospholipids in solution (10.1063/1.4948323; 10.1021/acs.jpclett.8b02241). Last research can be find on the following papers: doi:10.1021/ja900285z; 10.1021/jp511758w; 10.1038/srep28658; 10.1021/ja405993r; 10.1021/acschemneuro.7b00110.

REPLY: Although these observations are interesting, they are not related to PMLOs and therefore their discussion is not included to the manuscript.

Last, please discuss another interesting class of PLMOs molecules (some information can be found on DOI: 10.1016/j.bmc.2013.12.051; 10.1016/j.tibtech.2017.06.012)

REPLY: Although the mentioned papers describe structure and various applications of G-quadruplexes, they do not discuss G-quadruplexes as “another interesting class of PLMOs molecules”. There is not a single paper in PubMed that would discuss “G-quadruplex and liquid-liquid phase transition” or “G-quadruplex and liquid-liquid phase separation”, or “G-quadruplex membrane-less organelle”. Therefore, the proposed subject is unrelated to the topic of this perspective article.

Round 2

Reviewer 2 Report

The author has not satisfied any of the questions I have raised. Perhaps I was not clear enough. I would like to remind the author that this manuscript is a perspective and will be published in the Chemical-physical section. I have suggested some references that address the problem from Physical-Chemistry. In fact, the bibliography reported in the manuscript does not contain even one article pertinent to the Physical-Chemistry.
Polyamorphism is the capability of a material to exist in a different amorphous state. Many amorphous substances can exist in the different amorphous state (e.g. polymers, water, silica and so on). Polyamorphism requires two distinct amorphous states with a clear and discontinuous (Volume, Enthalpy, and Entropy) first-order phase transition. If a transition occurs between two stable liquid states, we will define a liquid-liquid phase transition.
All my suggestions are included in this liquid-liquid phase transitions.
Being the manuscript a perspective, the reader expects to find also suggestions of systems not currently studied in this view, but showing the characteristics treated in the manuscript (see G-Quadruplex).

REPLY1: I am not sure that the addition of this discussion is needed. In my view, this discussion is unrelated to the subject of this perspective.
Thermodynamics and chemical-equilibrium is the basic argument to treat any phase transition, please improve this topic.

REPLY2: Although formation of oligomers by amyloidogenic IDPs is not related to the subject of this article, discussion of the “aging” of PMLOs, their pathological transformation and potential relation of these processes to the pathogenesis of human diseases was added to the revised manuscript as a new section entitled “Dysregulation of the biogenesis of intracellular liquid droplets and disease”.
Not agree, in fact, the author discusses some question about apha-synuclein, an IDP. Are IAPP and Abeta IDP or not.
REPLY: Although these observations are interesting, they are not related to PMLOs and therefore their discussion is not included in the manuscript.
Not agree at all. See you my previous reply.
REPLY: Although the mentioned papers describe the structure and various applications of G-quadruplexes, they do not discuss G-quadruplexes as “another interesting class of PLMOs molecules”. There is not a single paper in PubMed that would discuss “G-quadruplex and liquid-liquid phase transition” or “G-quadruplex and liquid-liquid phase separation”, or “G-quadruplex membrane-less organelle”. Therefore, the proposed subject is unrelated to the topic of this perspective article.
I discuss this point in the first paragraph of this letter.